# Contrasting suitability and ambition in regional carbon mitigation

Yu Liu [1,2,12✉], Mingxi Du [3,4,12✉], Qi Cui [5,6,12], Jintai Lin [4✉], Yawen Liu[7], Qiuyu Liu[3,8], Dan Tong[9], Kuishuang Feng [10] & Klaus Hubacek [11]

Substantially enhancing carbon mitigation ambition is a crucial step towards achieving the Paris climate goal. Yet this attempt is hampered by poor knowledge on the potential cost and benefit of emission mitigation for each emitter. Here we use a global economic model to assess the mitigation costs for 27 major emitting countries and regions, and further contrast the costs against the potential benefits of mitigation valued as avoided social cost of carbon and the mitigation ambition of each region. We find a strong negative spatial correlation between cost and benefit of mitigating each ton of carbon dioxide. Meanwhile, the relative suitability of carbon mitigation, defined as the ratio of normalized benefit to normalized cost, also shows a considerable geographical mismatch with the mitigation ambition of emitters indicated in their first submitted nationally determined contributions. Our work provides important information to improve concerted climate action and formulate more efficient carbon mitigation strategies.

[1] Institutes of Science and Development, Chinese Academy of Sciences, Beijing 100190, China. [2] School of Public Policy and Management, University of Chinese Academy of Sciences, Beijing 100049, China. [3] School of Public Policy and Administration, Xi'an Jiaotong University, Xi'an 710049, China. [4] Laboratory for Climate and Ocean-Atmosphere Studies, Department of Atmospheric and Oceanic Sciences, School of Physics, Peking University, Beijing 100871, China. [5] School of Economics and Management, China University of Petroleum, Qingdao 266580, China. [6] School of Economics and Resource Management, Beijing Normal University, Beijing 100875, China. [7] Digital Economy Laboratory, University of International Business and Economics, Beijing 100029, China. [8] Department of Biological Sciences, University of Quebec at Montreal, Montreal, QC H3C 3P8, Canada. [9] Ministry of Education Key Laboratory for Earth System Modelling, Department of Earth System Science, Tsinghua University, Beijing 100084, China. [10] Department of Geographical Sciences, University of Maryland, College Park, MD 20742, USA. [11] Integrated Research on Energy, Environment and Society (IREES), Energy and Sustainability Research Institute Groningen (ESRIG), University of Groningen, Groningen 9747 AG, The Netherlands. [12] These authors contributed equally: Yu Liu, Mingxi Du, Qi Cui. ✉email: liuyu@casipm.ac.cn; dumingxi28@xjtu.edu.cn; linjt@pku.edu.cn

n December 2015, 195 countries approved the Paris Agreement aiming to limit the rise of global mean surface temperature to well below 2 °C above the pre-industrial level and to work towards 1.5 °C warming[1]. Each participating country agreed to submit its nationally determined contribution (NDC) every five years to report its emission mitigation ambition and implementation efforts. Whether the Paris climate goals can be achieved depends on the level of each country's climate mitigation ambition, which is in turn affected by that country's vulnerability to climate change, costs and affordability of mitigation, and other socioeconomic and political factors[2]. Mitigation efforts as per the first NDC are not sufficient to keep the temperature rise within 2 °C[3,4]. Although a recent study[5] indicates that fulfilling all conditional and unconditional pledges of updated NDCs for the whole world could keep the warming below 2 °C, there is still a certain distance to 1.5 °C warming. Straightening up information on economic costs and benefits of fulfilling such ambition could help with formulating mitigation strategies and enhance mitigation ambition. This is particularly true for major emitters due to large amounts of potential costs and benefits.

Past studies have estimated the costs of carbon dioxide ($CO_2$) emission reduction under the Shared Socio-economic Pathways (SSPs) based on integrated assessment models (IAMs)[6,7], and have used the carbon price, gross domestic product (GDP) loss or consumption loss as cost metrics[6,8–10]. For example, Yang et al. recently investigated mitigation benefits of NDCs using an IAM (the RICE model) and compared with emission based on solely economic emission level[7]. IAMs include a broad range of models with different economic mechanisms[11] important for assessing carbon mitigation costs. Several IAMs depict details of energy technology but include a simple economic module, by incorporating a growth function (e.g., in the REMIND model[12] and WITCH[13]) or using GDP as an exogenous input (e.g., in POLES[14]). Other IAMs are built based on computable general equilibrium models (CGEs), such as AIM/CGE[12] and EPPA[15]. These CGE-based models describe more realistic behaviours of economic agents, including producers, consumers, governments and investors, in response to price changes of goods and factors caused by carbon abatement. Although some models assume regions to be economically isolated (e.g., RICE[16]), in reality, one region's carbon reduction cost will be affected by other regions' mitigation actions through changes in international trade and capital flows. A number of global models have adopted the GTAP database to take into account the role of trade[11,15], including the GTAP-E CGE model[17–19].

Knowledge on the potential benefits, in addition to costs, of carbon mitigation is necessary to allow an understanding of the net economic effect. Here, the average reduction cost of carbon (RCC, US$ per $tCO_2$) is defined as the potential GDP loss in a given country/region as a result of action to remove one metric tonne of $CO_2$ emissions. The estimated average reduction cost of carbon can be contrasted against the potential benefits of emission reduction. The potential benefits of emission reduction are considered as the avoided economic costs associate with avoided climate damage, whose economic value is defined here to be equivalent to the social cost of carbon (SCC, US$ per $tCO_2$)[20–26] for each metric tonne of $CO_2$ emissions that can be otherwise removed. In particular, the method established by Ricke et al.[27] allows calculation of country-specific SCC based on climate model projections, empirical climate-driven economic damage estimation and socioeconomic projections. However, the average reduction cost of carbon has not been quantitatively compared with the SCC for all individual emitters of the world. This results in poor knowledge on the net economic effect (contrasting benefit and cost) of emission mitigation for many emitters, and thus on whether the mitigation ambition of a given emitter, relative to

others, is in line with the cross-regional ranking of the net effect for that emitter.

Here we contrast the average reduction cost of carbon against the potential benefit, valued to be equivalent to the SCC, for each of 27 countries or aggregated regions (Fig. 1), under 10 mitigation scenarios linked to the SSPs and Representative Concentration Pathways (RCPs). To derive the average reduction cost of carbon, emissions are assumed to be cut in 2020, for which year all data are available; this time choice is also consistent with the time horizon of SCC used here (2020 onwards). As detailed in Methods, we separate the individual major emitting countries and regions, such as China, the United States, the European Union, Japan, Russia, India, major participants in global climate negotiations. The amount of emissions removed under each scenario are defined as the difference in emissions between each scenario and SSP5-RCP8.5 (aka SSP5-Baseline), which is assumed to represent the highest emissions[28]. Under each scenario, the average projected emissions from a total of five IAMs (AIM/CGE, GCAM4, REMIND-MAGPIE, IMAGE and WITCH-GLOBIOM)[6] is used as the best estimate, with the range of emissions used as the uncertainty range. The RCC is calculated with GTAP-E[18,19] by implementing a carbon tax to achieve the emission reduction; and the SCC data are taken from Ricke et al.[27] The relative suitability for mitigation (RSM) is constructed as the ratio of normalized SCC to normalized RCC for each emitter. Considering the large uncertainties of the magnitude of mitigation cost and benefit but the general robustness of relative distribution, all the values of respective SCC and RCC have been normalized to range between 0 and 1 from the lowest to highest. Comparing the normalized values of RCC and SCC can better represent suitability (in terms of the benefit versus cost) among all regions and all scenarios. In sum, the normalization based on the min-max method is conducted to cancel out the effect of systematic errors (for all emitters) in the absolute values of RCC and SCC. We further contrast each emitter's RSM against its emission mitigation ambition, which is represented as the emitter's NDC-ambition score estimated based on its first NDC[2,29]. We find a large gap between the RSM and ambition of each emitter and offers insight to enhance mitigation ambition through improvement of international cooperation with mutual economic benefits.

## Results

**Reduction cost of carbon.** Figure 2 shows the spatial distribution of average reduction costs of carbon under each mitigation scenario. Results are shown for 23 scenarios, including the 10 scenarios with available SCC results and the other 13 scenarios for completeness. The global RCC ranges from US$16.5 per $tCO_2$ (15.4–17.7) for Scenario SSP3-Baseline to 45.8 per $tCO_2$ (37.7–62.4) for SSP1-RCP1.9. Scenarios with higher mitigation targets tend to have higher global average RCC values. This is because higher emissions mitigation will lead to a higher marginal mitigation cost[30–32]. In GTAP-E, the emission mitigation through carbon tax raises the cost of fossil fuel use and thus reduces the industrial production and GDP. A larger carbon abatement requires a higher carbon tax and thus a greater reduction in production for cutting one unit of emission.

Figure 2 shows that under almost every scenario, the European Union and Switzerland have the highest values of RCC whereas Thailand and Rest of Western Asia have the lowest RCC. Taking Scenario SSP2-RCP4.5 (which roughly represents middle of the road) as an example, the RCC values of the European Union and Switzerland are US$ 97.2 (89.7–114.7) per $tCO_2$ and US$ 93.3 per $tCO_2$ (86.2–111.2), respectively, which are more than five times that of the United States at US$ 18.4 per $tCO_2$ (17.1–21.5). Regions such as Thailand (US$ −14.0 per

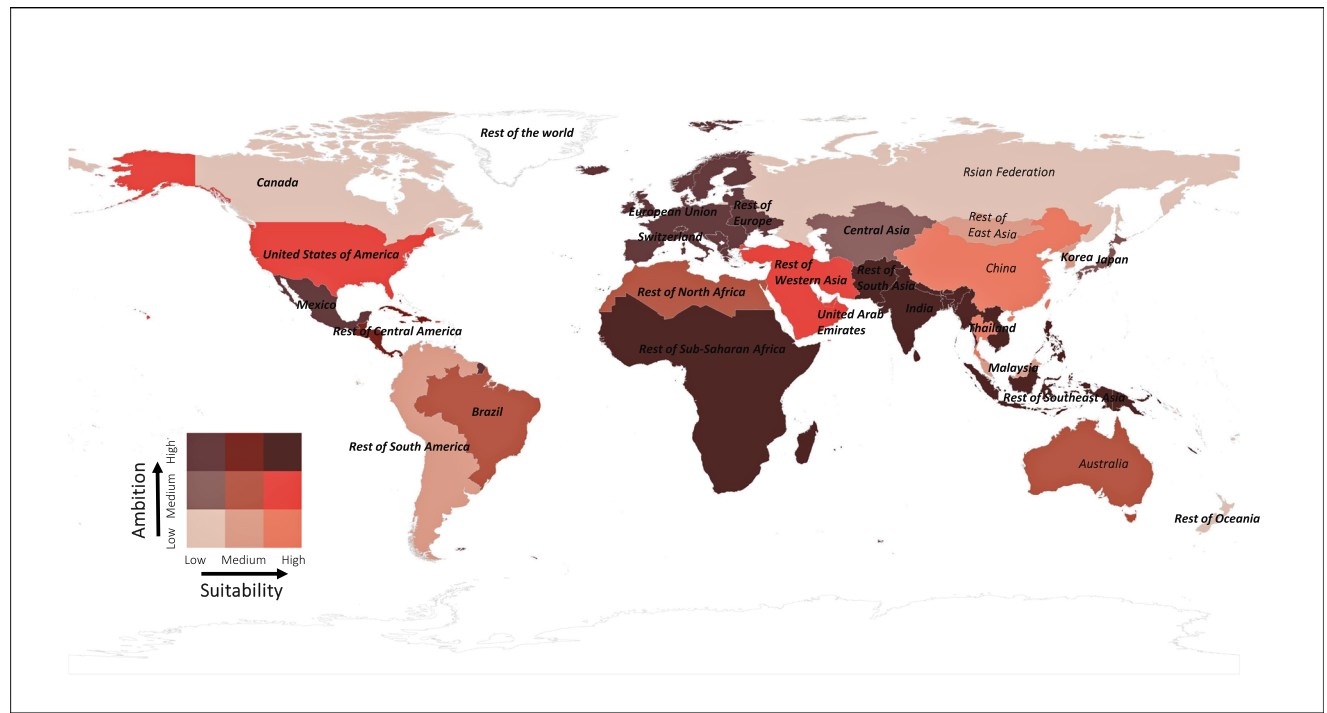

**Fig. 1 Contrasting suitability and ambition of carbon mitigation among 27 emitting countries and regions.** The RSM (relative suitability of mitigation) and NDC (national determined contribution) ambition for each region under SSP2-RCP4.5. For both RSM and ambition scores, "High" represents the top 1/ 3 among the 27 regions, "Medium" represents the middle 1/3, and "Low" represents the bottom 1/3.

tCO$_2$ (−19.0 to −10.4)), Rest of Western Asia (US\$ −4.5 per tCO$_2$ (−25.4 to 3.5)) and China (US\$ −0.8 per tCO$_2$ (−2.6 to 0.9)) would even obtain negative RCC values under SSP2-RCP4.5 — in other words, these regions would obtain an economic gain from emission reduction. This is because when all countries act to reduce emissions and raise the cost of production worldwide, these countries with lower reduction costs would have smaller declines of capital rent, and attract more capital inflow for investment (Supplementary Fig. 2).

The inter-regional difference in average reduction cost of carbon is largely determined by the difference in mitigation marginal cost, which is reflected in the carbon tax rate and is highly dependent on energy consumption mix and energy intensity, across the countries/regions. In particular, the countries with higher proportions of fossil fuels in their energy consumption mix require lower carbon tax rates to achieve the same percentage of carbon abatement, because for them fossil fuels can be substituted by non-fossil energy at relatively low costs. Examples of these countries include Malaysia, Thailand, Korea, Rest of Southeast Asia and Mexico, for which more than 80% of energy consumption is supplied by fossil fuels. In addition, the countries with higher energy intensities tend to have lower carbon tax rates, because they can reduce energy intensities at lower costs. As China, India and the United States have higher energy intensities than the European Union and Switzerland, they have much lower mitigation marginal costs than the latter two emitters.

In addition, we supplemented the sensitivity experiment to explain the impact of inter-regional transmission and feedback mechanism on the average reduction cost of carbon (Supplementary Data 5). In the test, we turned off the price transmission mechanism of GTAP-E model, making each region as a single-regional CGE model. In this case, the average reduction cost of carbon in each region is not affected by the emissions reduction of other regions. The results of sensitivity experiment show that if the mechanism of inter-regional transmission and feedback is ignored, the economic cost associated with average reduction cost of carbon will be overestimated or underestimated. In sum, the effect of trade mechanism could be mainly explained by two major channels. On one hand, the global carbon mitigation will directly reduce energy demand and energy price to a certain degree. When the trade mechanism is motivated in GTAP-E model, energy-importing regions will benefit from the lower energy price and cut down the costs of their domestic production, alleviating the GDP losses caused by carbon mitigation. But energy-exporting regions will experience greater GDP losses because of the decreasing revenues from energy exports. On the other hand, as energy-exporting countries mostly have a relatively high carbon intensity, carbon mitigation will raise their costs of domestic production more significantly, compared with energy-importing countries. As a result, energy-exporting countries will lose the comparative competitiveness in the global market, aggravating GDP losses caused by carbon mitigation.

Comparing the regional RCC and SCC further shows that emitters with higher SCC tend to have lower RCC for all 10 scenarios with both RCC and SCC results available (Fig. 3 and Supplementary Fig. 3). This contrast is associated with the geographical (latitudinal) distribution of economies. As explained above, the inter-regional inequality in RCC is mainly due to the differences in energy consumption mix and energy intensity, which leads to the RCC being generally higher in developed regions than in developing regions. In contrast, the SCC tends to be lower at high latitudes, where developed regions are mainly located, and higher at low latitudes, where many developing regions are located[27,33]. Although all countries would suffer great losses as warming is aggravated in the long term, countries at high latitudes tend to suffer less (and may even gain) from warming in the short run[8,33].

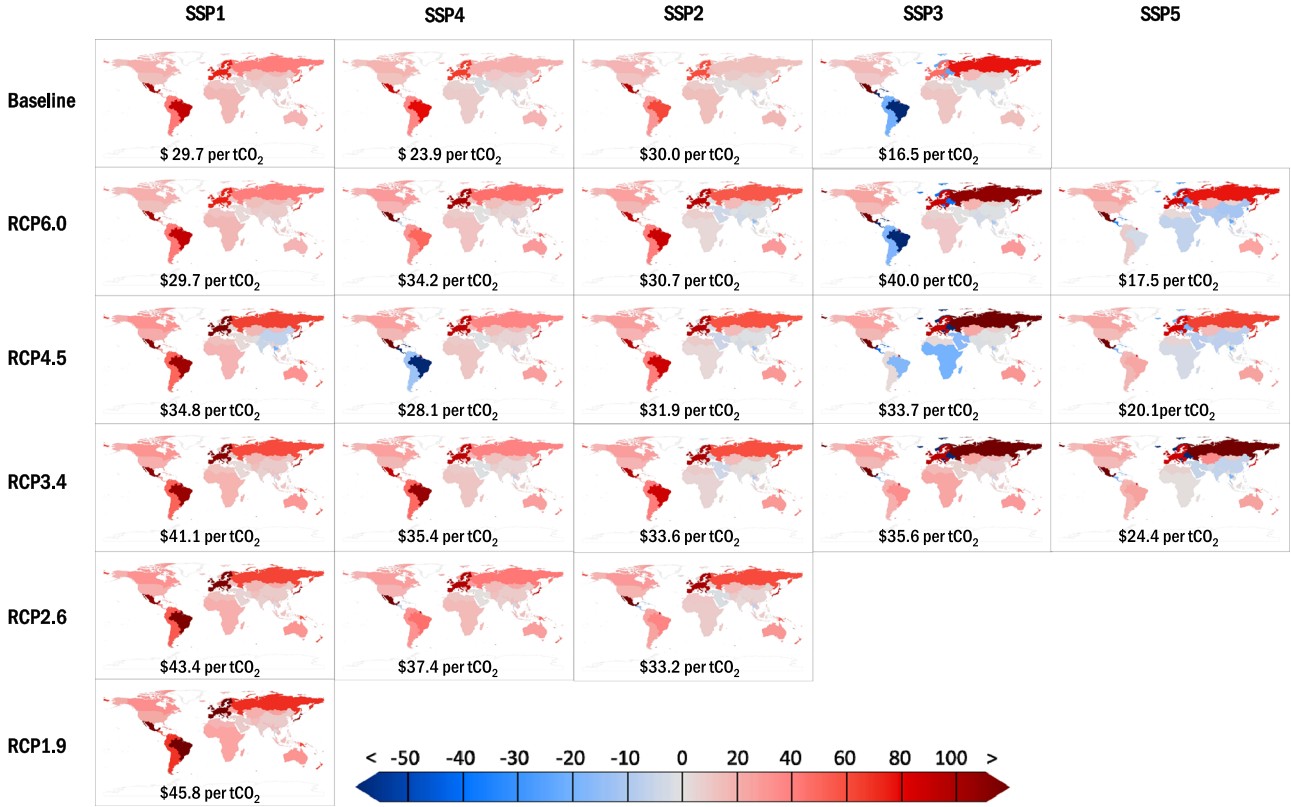

**Fig. 2 Substantial cross-regional disparity in cost of carbon mitigation.** The RCC (average reduction cost of carbon) for each region under each scenario (unit: US$ per $tCO_2$). Red color denotes regions with positive value of RCC, and blue color represents regions with negative value of RCC. Detailed value of RCC for each region under each scenario can be found in Supplementary Data 3.

**Relative suitability of mitigation**. We further evaluate the RSM results to examine which emitters are more suitable to cut emissions based on a cost-benefit analysis. Figure 4 shows that under the 10 mitigation scenarios, Rest of Western Asia (1.5–3.5, depending on the scenario) and India (1.2–4.4) have the highest values of RSM, and the European Union have the lowest (0.01–0.04). The RSM values for China and the United States are much larger than for the European Union, with values of respectively 0.73, 0.97 and 0.02 in Scenario SSP2-RCP4.5 (Fig. 4). For many emitters, the values of RSM do not change significantly across the 10 mitigation scenarios. However, there are instances when the RSM of an emitter under one scenario substantially deviates from the RSM values under other scenarios, mainly as a result of a small percentage reduction in emissions leading to a small value of average reduction cost of carbon. For example, under SSP3-RCP6.0, Brazil only needs to cut emissions by 0.6%, as compared to a global average mitigation of 4.3%. Detailed RSM results for each region under each scenario can be found in Supplementary Data 4.

Contrasting the cross-regional ranking of RCC and RSM is useful, given that mitigation cost is often proposed as a key parameter to guide regional mitigation[34]. We find a significant mismatch between the ranking of RSM and RCC under every mitigation scenario (Supplementary Data 4). Under Scenario SSP2-RCP4.5, the RSM of Korea ranks the 16th highest, in contrast to its RCC ranking at the 3th lowest; and the RSM of the United States ranks the 3th highest, in contrast to its RCC ranking at the 16rd lowest (Supplementary Fig. 4). The contrast in ranking between RSM and RCC suggests that considering both costs and benefits of emission mitigation of each emitter (relative to other emitters) would provide more complete information for

determining regional mitigation ambition to achieve global emission reduction.

**RSM and NDC ambition**. Figure 1 contrasts the RSM and NDC-ambition score[2,29] of each emitter under Scenario SSP2-RCP4.5. Results for other mitigation scenarios are similar (Supplementary Data 4). Although China, Rest of Western Asia, Rest of South-East Asia and the United States are relatively suitable mitigation regions with high values of RSM (ranked in top 1/3), their NDCs do not show correspondingly strong mitigation ambition (ranked in bottom 1/3). These emitters could enhance their ambition not only because they are often major emitters but also because they are economically more suitable regions of carbon mitigation (compared to other regions). China's political leadership has announced their intention to become carbon neutral before 2060[35,36], and it is expected that the country will announce substantially strengthened emission mitigation ambition in its next NDC. The United States has re-joined the Paris Agreement and even enhanced their ambition under the newly elected political leadership[37]. The enhanced ambition of these top two emitting countries would be very important for boosting global climate action to levels consistent with the Paris goal.

Figure 1 shows that Rest of East Asia and Rest of South America rank medium (middle 1/3) in RSM but low in mitigation ambition (bottom 1/3). These developing countries might need external financial and technical aids through international collaborations to enhance their affordability, capability and thus ambition of carbon mitigation. Figure 1 also shows that the European Union and Switzerland have relatively low values of RSM but are among the most ambitious regions in emission mitigation. These developed countries could consider to help the

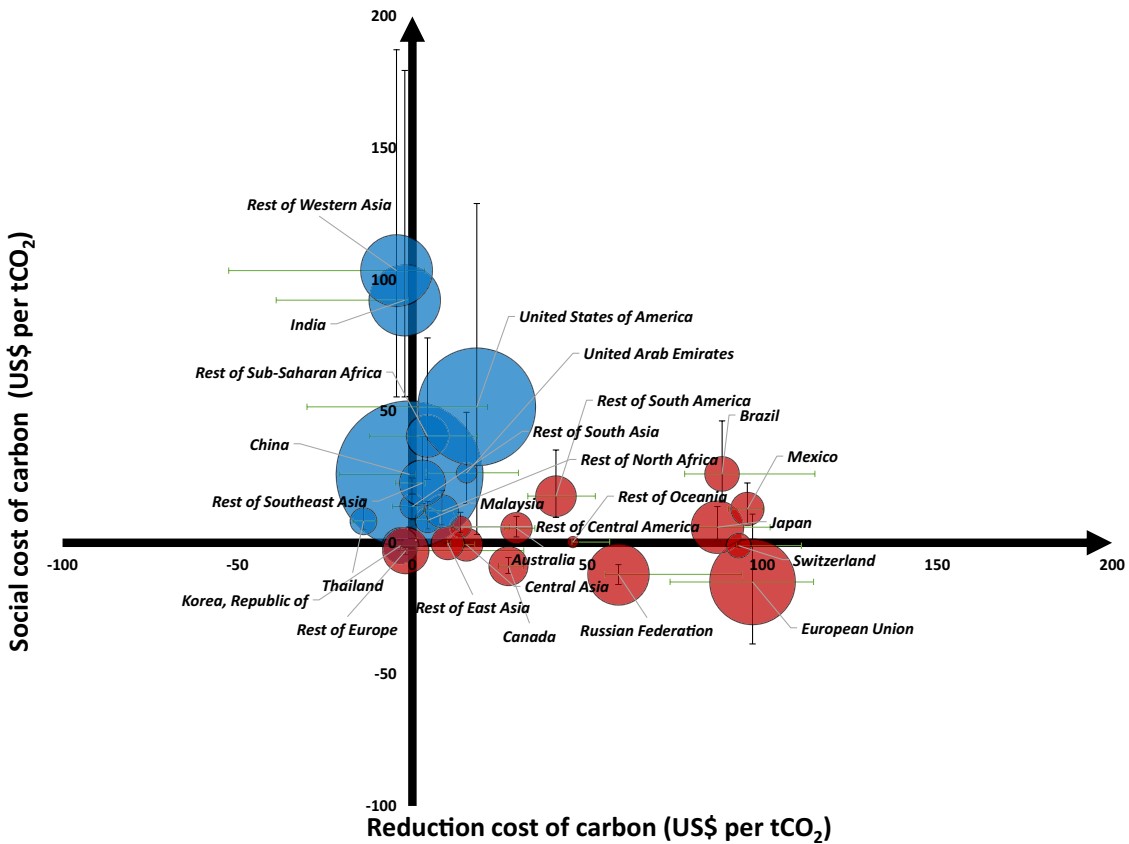

**Fig. 3 Contrast between costs and benefits of carbon mitigation for individual emitters.** The RCC (average reduction cost of carbon) and SCC (social cost of carbon) for each country/region under SSP2-RCP4.5. The size of the dots denotes the magnitude of regions' CO₂ emissions. Regions with higher SCC than RCC are shown with the blue color, and regions with higher RCC than SCC are shown with red color. The vertical error bar represents the 66% CI of SCC. The horizontal error bar represents the maximum and minimum RCC values based on results from five IAM models.

aforementioned developing regions with higher RSM but lowest ambition to reduce carbon emissions through financial and technical mechanisms. This would boost the world's ambition and action as a whole to mitigate climate change at lower costs than if individual, uncoordinated actions are taken. Several developed countries and states have attempted to link their local carbon markets, aiming to reduce carbon emissions through capital and technology transfer. For example, California and Québec have linked their Emissions Trading Systems in 2014, and the European carbon market keeps expanding[38]. Besides, the idea of forming a "climate club" to achieve strong international mitigation cooperation among the major economies has also been promoted in recent years[39,40].

Considering that the average reduction cost of carbon of each region is not only affected by one region's mitigation target but also other emitters' actions, it is important to understand how globally concerted mitigation action affects the RCC values for individual emitters. Here we employ a decomposition analysis approach[41,42] to quantify the effects of other regions' mitigation actions on a given emitter's average reduction cost of carbon. Our results show that the RCC values of European Union and United States are about 13 and 17% lower when other regions also cut emissions under Scenario SSP2-RCP4.5. The effects are even stronger for many regions like China and Russian Federation (Supplementary Fig. 5 and Supplementary Data 5).

To further demonstrate the economic mutual benefits of cross-regional emission mitigation collaboration, we conduct a hypothetical experiment based on Scenario SSP2-RCP4.5 with GTAP-E model, in which European Union, a region with low RSM and high ambition, transfers 10% of their mitigation amount (30.4 (28.3–40.6)

million tCO₂) to China, a region with high RSM and low ambition (although with a recent ambitious pledge for carbon neutrality[35,36]). We find that the RCC of the European Union would decrease significantly from US$ 97.2 per tCO₂ (89.7–114.7) to US$ 90.1 per tCO₂ (82.5–105.7). For China, its average reduction cost of carbon would only increase slightly from US$ −0.8 per tCO₂ (−2.6 to 0.9) to US$ 0.9 per tCO₂ (−0.7 to 2.2). Therefore, a win-win situation could be achieved through the Sustainable Development Mechanism[1], with necessary improvements, to support China's carbon mitigation. This Sino-Europe collaboration would also avoid US$ 4.65 (3.55–7.37) billions of GDP loss for the world through trade-associated inter-regional connections. If the transferred portion of emission mitigation increases to 50%, the avoided world GDP loss would increase to US$ 20.2 (15.8–31.2) billion; and the average reduction cost of carbon would become US$ 5.9 per tCO₂ (4.5–6.9) for China and US$ 52.1 per tCO₂ (42.0–64.3) for the European Union.

## Discussion
Our study is subject to a few uncertainties and limitations. First, the emissions under each scenario are averaged over simulation results from five IAM models. Although SSP scenarios are not meant to directly represent the real world, they are being considered as investigating different possible futures. In other words, the real world would be covered among all SSP scenarios. As is shown in Supplementary Fig. 3a, our findings are robust not only under this SSP2-RCP4.5 scenario, but also under all possible mitigation scenarios from SSP1-5 and RCP4.5-Baselines. The comparison of RSM and ambition under each mitigation scenario is provided in

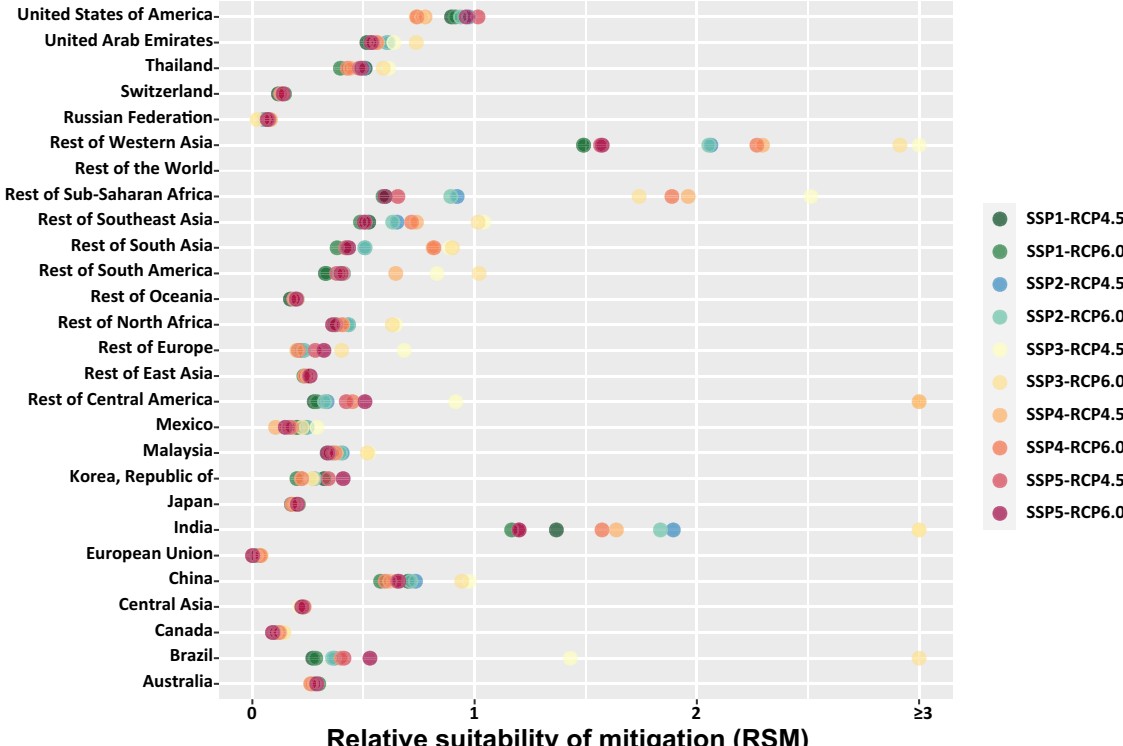

**Fig. 4 Substantial cross-regional disparity in suitability of carbon mitigation.** RSM (relative suitability of mitigation) for each region under each scenario. RSM is defined as the ratio of normalized SCC (the social cost of carbon) to normalized RCC (the average reduction cost of carbon). Detailed values of RSM for each emitter under each scenario can be found in Supplementary Data 4.

Supplementary Fig. 7. Although the accuracy of each model is subject to errors in model parameters and assumptions[6], the multi-model averaging reduces the influence of errors in individual models. We also provided the comparison of SCC and RCC based on each IAM model in Supplementary Fig. 3b–f.

Second, we use SSP5-RCP8.5 to be the scenario with the highest emissions, relative to which we calculate the emission reductions under other scenarios. Exceptions occur under SSP3-Basline (which is not used in evaluating RSM), under which three emitters have emissions higher than under SSP5-RCP8.5. In this case, we assume there is zero emission mitigation for that emitter. Third, the calculation of SCC follows Ricke et al.[27], and is affected by the statistical method and functional forms used to assess economic damage, although the relative ranking of countries is robust under each scenario[33]. We have further tested the robust of our results by constructing RSM with SCC results by Yang et al.[43] based on the latest version of the RICE model with modifications and updates on the climate module, regional definition and damage function. Our findings about the mismatch between suitability and ambition (Supplementary Fig. 6 and Supplementary Data 5) is still robust and significant even with different kinds of SCC, because what behind this phenomenon is the negative correlation between the global economic landscape and geographic distribution.

Fourth, the GTAP-E model used to calculate the RCC is a static economic model, which calculates the changes in individual economies from one equilibrium to another without explicitly specifying the path of economic evolution. Fifth, our calculation of RCC is done for 2020, which is consistent with the time horizon of SCC (2020 onwards). Our additional calculation of RCC for 2030 also shows cross-regional ranking similar to that for 2020 (Supplementary Data 4). Sixth, the allocation of carbon reduction across countries could be determined by optimizing a global social welfare function and achieving the Lindal

equilibrium theoretically[44,45], but the carbon reduction used here is determined according to each country's cost and benefit, which is regarded as a Nash equilibrium[46]. Therefore, although the carbon reduction based on the RSM index could achieve the social welfare optimum of each country, it is unable to ensure that the global welfare optimization is reached. However, it could still be regarded as a second-best scheme of carbon reduction because the global welfare is improved from the current NDC mechanism. In sum, although the uncertainties in the absolute values of both RCC and SCC are large for individual emitters, the negative spatial correlation between RCC and SCC is consistent across the mitigation scenarios (Supplementary Fig. 3), so is the cross-regional ranking of RSM (Supplementary Data 4).

How to improve the international cooperation rather than individual policies on climate change mitigation is crucial to achieving the Paris goal[47]. Although a cross-regional policy framework has been proved as an effective solution considering large differences across regions to reach net-zero carbon emissions[48], how to build the cooperation for regions with cross-regional policy framework is the major challenge. Besides, another major challenge is large disagreement about the benchmarks by which each country's should enhance their ambition[49]. This study offers an RSM-based framework to help raise regional emission mitigation ambition and provides a guidance for mitigation cooperation from the economic perspective. More affordable emitters with low RSM but high ambition, particularly the European Union and Switzerland, and less affordable developing countries with high RSM but low ambition might consider working collaboratively to reduce emissions and share credit of such action. Such cooperation would be more economically viable for both parties and is supported by the 6th Article of the Paris Agreement. For example, the cooperation between Norway and Indonesia based on reducing carbon emissions from deforestation and forest degradation has already shown some contributions to

tropical countries NDCs and emissions reduction[50]. Together, enhancement of domestic and internationally collaborative mitigation action, aided by better knowledge on cost and benefit and thus enhanced ambition, will be crucial for successful climate change mitigation.

## Methods

**Region and scenario setting**. We separate the world into 27 countries and aggregated regions based on economic volume and geographical location, similar to our previous study[51,52]. These regions are detailed in Supplementary Data 1. We obtain the emission data for different scenarios from the SSP database (https://tntcat.iiasa.ac.at/SspDb/)[6,53,54]. The scenarios with a brief methodology framework are specified in Supplementary Fig. 1. The SSP database includes 5 groups: OECD (the OECD 90 countries and the European Union member states and candidates), REF (the reforming economies of Eastern Europe and the Former Soviet Union), ASIA (Asian countries except the Middle East, Japan and the Former Soviet Union states), MAF (the Middle East and Africa), and LAM (Latin America and the Caribbean).

Then, we calculate the mitigation target in the year of 2020 for each group under each scenario as the relative difference in emissions between SSP5-RCP8.5 (which is assumed to represent the highest emissions[28]) and that scenario.

$$M_{s,g} = (E_{s',g} - E_{s,g})/E_{s',g} \qquad (1)$$

Here, M and E denote the mitigation target and emission, respectively. The subscript s denotes a scenario, s' denotes the reference scenario SSP5-RCP8.5, and g denotes each of the 5 groups (OECD, REF, ASIA, MAF and LAM). $M_{s,g}$ is set to be zero when $E_{s,g}$ is greater than $E_{s',g}$, which situation only occurs under SSP3-Baseline for which three emitters (Rest of Europe, Central Asia and Russia Federation) have emissions higher than under SSP5-RCP8.5 by 10%.

Subsequently, the mitigation target of each of the 27 country/region ($M_{s,r}$) is set to be the same as the target of the group ($M_{s,g}$) to which that country/region belongs. Detailed results of the mitigation targets are shown in Supplementary Data 2.

**RCC calculation**. We calculate the RCC with the GTAP-E model[18–19]. For each scenario, the economic effect of emission reduction is simulated in GTAP-E by implementing carbon tax at a level consistent with the emission mitigation target in 2020. All RCC values are expressed in 2014 constant price. Detailed results of RCC can be found in Supplementary Data 3.

The GTAP-E model is a multi-regional, multi-sector economic equilibrium model, developed based on the GTAP model. As a comparative static analysis model, GTAP-E assumes that the returns to scale of production remain unchanged in the completely competitive market; and producers maximize the profits while consumers maximize the utility. The equilibrium of total supply and demand determines the values of endogenous variables, such as commodities prices, wages, capital return, and land rents. All economies (countries and regions) connect with each other through commodity trade.

GTAP-E includes three representative agents, that are producers, private households, and governments. The activity of producers is described by a sequence of nested constant elasticity of substitution (CES) functions, which aim to reproduce the substitution possibilities across the full set of inputs. On the top level, the total input is composed of two aggregate composite bundles, i.e., intermediate demand and value added. The second level nest decomposes each of the two aggregate composite bundles into their components, such that one is demand for individual intermediate goods and the other is demand for primary factors. The final nest accepts the Armington assumption to allow an incomplete substitution between domestically produced goods and imported goods.

Built upon GTAP, GTAP-E improves the modelling of energy input structure, carbon dioxide emission, and mitigation policy. (1) A new nesting structure of energy commodities is introduced into the bundle of primary factors. The energy composite is combined with capital to produce an energy-capital composite, which is in turn combined with other primary factors in a value-added-energy (VAE) nest through a CES structure. The energy composite comprises electricity and non-electricity energy. The non-electricity energy is composited by coal and non-coal commodities, with non-coal further composited by gas, oil, and petroleum products. (2) The carbon dioxide emission is also introduced, accounting for the emission from the burning of fossil fuels by production sectors and households. The carbon dioxide emission factors of fossil fuels are derived from Vermeulen (2014)[55]. (3) The regional real carbon tax is developed, defined as the nominal tax rate deflated by the income disposition price index. The carbon tax could be employed to achieve the goal of carbon abatement, by reducing the utilization of fossil fuels in production sectors and households.

The consumption preferences of private households are represented by the constant differences of elasticities implicit additive expenditure function by Hanoch[56]. The Cobb–Douglas function is adopted to represent government consumption. The aggregate volume of investment comes from the identity that the nominal investment equals saving, where saving is the sum of domestic saving and net capital inflows from foreign economies. Investment expenditures on the composite goods are described by a Leontief utility function, and subsequently decomposed into demand for domestic and imported goods.

Within each economy, the GTAP-E model allows capital and labour to move between production sectors, and partially allows land to move between crop producing sectors. The full employment of labour is assumed. The savings of regions are pooled to the global investment, and the latter is allocated to different regions according to their return of capital.

The latest version (v10a) of the GTAP database is utilized, which is constructed from the input–output tables of 141 countries and regions across the world with a base year of 2014[57]. The GTAP database contains 65 sectors and 5 primary production factors. For this study, the 141 countries and regions have been aggregated to 27 regions (Supplementary Data 1), which specify major producers, consumers, and importers/exporters. The 65 production sectors are aggregated to a total of 8 sectors (Supplementary Data 1).

**SCC and RSM calculations**. The RSM for each scenario and country/region is defined as follows:

$$RSM_{s,r} = nSCC_{s,r}/nRCC_{s,r} \qquad (2)$$

Here, the subscripts s and r denote the scenario and country/region, respectively. For each scenario and region, nSCC and nRCC are the values of respective SCC and RCC normalized with the Min-max method, and thus range between 0 and 1. The RCC is calculated by GTAP-E. The SCC for the 27 regions are mapped from the country-level SCC (cSCC, for 2020 onwards) data from Ricke et al.[27].

As detailed in Ricke et al.[27], the cSCC are calculated in several steps. First, the GDP growth rates are calculated based on the GDP and population assumptions in the SSPs[6]. Second, the magnitude and geographic pattern of temperature change under different RCPs, the carbon cycle and the climate system responses are obtained from climate models[58–61]. Third, damage modules are used to convert country-level temperature and precipitation changes into country-level economic damages[62,63]. Finally, the time series of future damage is converted to the present value of cSCC with a discounting module[64,65]. Following Ricke et al.[27], we adopt the cSCC data computed by the central specification of the Burke–Hsiang–Miguel (BHM) damage function (short run, no income differentiation) and a growth adjusted discount rate ($\rho = 2\%$, $\mu = 1.5$). The values of cSCC are converted to 2014 constant price in this study.

Supplementary Data 4 presents results of SCC, RCC, nSCC, nRCC and RSM for each scenario and country/region.

**NDC ambition score**. We use the NDC ambition score of each country from Robiou du Pont and Meinshausen (2018)[29], which was calculated based on the country's first released NDC. Tørstad et al[2] also used this score to discuss regional emission mitigation ambition. The score was determined based on the degree of warming, ranging from 1.2 °C (most ambitious) to above 5.1 °C (least ambitious), had the NDC of a given country been applied globally.

According to the ambition scores, we classify the regions into three categories: high ambition (top 1/3), medium ambition (middle 1/3), and low ambition (bottom 1/3). For 8 aggregated regions (including Central Asia, Rest of Central America, Rest of Europe, Rest of North America, Rest of South America, Rest of Southeast Asia, Rest of Sub-Saharan Africa, and Rest of Western Asia) containing countries with different ambition scores, we take the emission weighted score to represent the ambition of that region. For United Arab Emirates, the score is considered as the same as Rest of Western Asia. Detailed results of ambition scores are shown in Supplementary Data 1.

**Reporting summary**. Further information on research design is available in the Nature Research Reporting Summary linked to this article.

## Data availability

Source data are provided with this paper. All data used here are cited in the text or provided in the supplementary files.

## Code availability

All computer codes generated during this study are available from the corresponding authors upon reasonable request.

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

## Acknowledgements

This study is supported by the National Natural Science Foundation of China (72125010 (Y.L.), 71974186 (Y.L.), 42075175 (J.L.), 71903014 (Q.C.)), the second Tibetan Plateau Scientific Expedition and Research Program (2019QZKK0604 (J.L.)), and the Young Talent Program of Xi'an Jiaotong University (GG6J007 (M.D.)).

## Author contributions

Y.L., M.D., and J.L. conceived the research. Y.L., M.D., Q.C. and J.L. designed the research. Q.C., M.D. and Y.L. performed the research with inputs from Yawen.L., Q.L. and D.T. M.D., Q.C. Y.L. and J.L. analysed the results. M.D., J.L., Q.C. and Y.L. led the writing with inputs from K.H. and K.F. Y.L., M.D., Q.C., J.L., Yawen.L., Q.L., D.T., K.F., and K.H. discussed the results and commented on the manuscript.

## Competing interests

The authors declare no competing interests.
