## [Peer Review File · Nature Communications]

Peer review comments, first round review –

Reviewer #1 (Remarks to the Author):

This research creatively utilizes a simple yet effective indicator, the RSM, to answer an important question: how we should conduct carbon emission mitigation in the most cost-effective way under the international scope. The research question is in accordance with the “common but differentiated responsibilities” principle proposed by UNFCCC, thus bearing great significance in policy suggestions for international climate mitigation collaborations and negotiations. The manuscript itself is very well structured. The presentation of method is clearly organized and articulated, nicely facilitated with schematic diagram for readers’ quick understanding. Results are effectively illustrated in the figures produced, accompanied by insightful analysis to formulate an answer to the research question proposed in the beginning. Overall, this research manuscript is produced in very high quality. I believe it is almost ready to be scientifically published.

One comment regards to the terminology. In the entire discussion, the authors seem to associate RCC only with carbon tax. However, it might be a too narrow definition for carbon pricing tools. The other most important tool for carbon pricing is emission trading scheme (ETS). In addition to setting a fixed tax rate for each tonne of carbon emitted, the same carbon price could also be achieved through “cap and trade” schemes. Hence, narrowing the discussion for carbon pricing to only carbon tax would not be the most precise approach. I would suggest the authors to consider broadening their discussion on carbon pricing to wider variety of policy tools.

Please find the following my comments:

1. After reading the whole paper, I can understand your title, but it is not that straightforward what your research is about for a broader audience. It would be nice to rephrase it or clarify the regional carbon mitigation “suitability”.
2. Intended Nationally Determined Contribution is usually abbreviated to INDC.
3. Authors used GTAP-E model in this work. Many other IAMs, such as DRCE/RICE, have also been used to discuss the similar topics (Yang et al., 2018, Mi et al., 2019). It will be better if authors add more explanations why GTAP-E was selected. What are its advantages?
4. Line 47: Better define the economic benefits here. I’m not sure if the “incomplete and uncertain information on the economic costs and benefits” are the primary barriers to promoting enhanced mitigation ambition. Every policy in the US needs to pass a cost-benefit test, yet they have no invitation to do as climate change is a public good.
5. Line 60-63: I don’t think international trade will affect the low-carbon technology cost for countries. Could you provide an example of how carbon reduction costs will be affected? Adding a definition for the carbon reduction cost can also be very helpful.
6. “The cost of carbon emission reduction” is an essential concept in this paper and I suggest using words rather than abbreviations to provide a fluent reading experience. Also, as the unit is US\$/tCO₂, should you use ‘unit cost of emission reduction’ or similar ones to avoid misunderstanding?
7. Reference 7 also investigates the benefit aspect of NDCs. How is your research different from theirs?
8. Line 68-71: The sentence is quite hard to read. What are the potential benefits of emission reduction? Add an example here. Reference 18-24 adopted different equations to calculate SCC. Some defined the SCC as welfare loss, while others define SCC as GDP loss. The SCC estimation is blooming in recent years. I suggest the authors refer to more updated SCC estimates under SSPs

and justify why you are choosing Ricke's estimates.

9. Figure 1: Better avoid abbreviation in figure titles.

10. Figure 3: Why do the rest of Western Asia and the US have such a huge uncertainty range for SCC?

11. Figure 4: How to interpret these RSM numbers? Where is the benchmark for domestic mitigation/international mitigation transfer?

12. I suggest the authors separate the conclusion from the discussion and be more explicit about the findings and policy implications.

References:

P Yang, YF Yao, Z Mi, YF Cao, H Liao, BY Yu, QM Liang, DM Coffman, YM Wei. Social cost of carbon under shared socioeconomic pathways. *Global Environmental Change*. 2018, 53: 225-32.
Z Mi, H Liao, D Coffman, YM Wei. Assessment of equity principles for international climate policy based on an integrated assessment model. *Natural Hazards*. 2019, 95, 309-323.

Reviewer #2 (Remarks to the Author):

This is a great paper and should be published by Nature Communications. I do not have any important reservations and so I recommend simple acceptance, but if other reviewers raise any important technical questions I urge the editors to give the authors generous opportunity to respond.

Mindful of the impending IPCC deadline, if there are no objections that are legitimate raised by other reviewers, I urge the journal to consider a rapid acceptance.

This paper provides a novel comparison, for 27 nations, of the costs of mitigation to the benefits of avoided damages under policy-relevant mitigation scenarios for each of those nations, and to ambition in NDCs. The authors choose logical and defensible methods for doing this among those that are readily available, and explain and write their paper with admirable clarity -- clarity that far surpasses the average climate policy article in journals like Nature and Nature Climate Change. As with any such exercise, there are many ways in which a reader might quibble with one or more aspect of the chosen methods (and in particular, the models relied upon, and their parameter assumptions) -- but that is unavoidable, and the authors are forthright about this and the assumptions made, and again the authors have chosen logical, defensible, and policy-relevant methods currently available to make these important comparisons.

Mainly for this reason, I believe the paper should be published in this journal, and if any referees disagree based on quibbles about the methods, I urge the editors to critically examine whether such quibbles actually point to technical worries about the implementation of those methods, or to important worries about correct choice of methods, since again any possible choice of methods will be subject to quibbles, and the authors seem to me to have made reasonable choices for deriving a 'central readily-available estimate' for these comparisons.

With the comparisons in hand, the authors wisely connect the dots between their results and how they can help inform pressing policy questions, such as how to form effective 'linkages', 'climate clubs', where to expect barriers to arise to global cooperation on GHG reductions, etc. They connect the dots to the underlying conceptual and empirical considerations: eg that what matters for mitigation opportunities is partly a matter of where it is cheap to make reductions, not merely where there are a lot of emissions, etc. The authors also provide clear and well-designed supplemental data summarizing the quantitative results and the numbers used to derive them that are themselves the outputs of models or other studies.

One idea, perhaps for the authors' future work, is to test the robustness of their central results to

the use of other alternative models. For example, one could look at SCC values using other widely used multi-region SCC models, eg RICE, FUND, or PAGE. (Note that the Ricke damage function used by the authors is not used by eg US government estimates, and is based on controversial Burke-Hsiang-Miguel damage function.) Similarly, the authors could try alternative mitigation cost models (eg GCAM). The authors could test robustness for the subset of their 27 regions for which there is a direct comparison with regions within a chosen alternative SCC model and chosen alternative mitigation cost model. Because I assume that the basic qualitative story would remain the same given different model choices -- ie the same nations would have qualitatively similar match or mismatch between costs and benefits and ambition -- I do not demand that the authors perform these robustness checks, but it might add value, perhaps in future work. The authors could perhaps also add a sentence explaining the normalization method and the motivation for it just a slight bit more.

The paper will be the basis for useful future work as well that builds on this -- eg evaluation of climate club possibilities.

Again, this is a great paper and should be published by Nature Communications. I do not have any important reservations and so I recommend simple acceptance, but if other reviewers raise any important technical questions I urge the editors to give the authors generous opportunity to respond.

Thanks to the authors for this contribution.

Mark Budolfson

Reviewer #3 (Remarks to the Author):

The study develops the relative suitability for mitigation (RSM), as a metric to compare countries' relative potential for further mitigation at a number of SSP/RCP scenarios. For each country the RSM is the ratio of the local social cost of carbon (SCC, defined as the discounted sum of marginal damages to the country from a ton of emitted CO₂) to the local marginal cost of reducing a ton of emissions at the respective scenario. The local SCC is taken from the literature (Ricke et al) and the marginal cost is computed by implementing the particular scenarios into GTAP E, an integrated assessment model (IAM) with the distinction of capturing trade effects.

It is important to understand that the metric evaluates the suitability for FURTHER abatement beyond the particular scenario in question. The evaluation does not depend on current emissions or even on the nationally determined contributions.

The authors find that there is large variation in RSM across regions and countries, and suggest countries with high RSM might have incentives to do more than under the given scenarios and those with low RSM might choose to trade mitigation effort with high RSM countries (they suggest the EU exchange it's reductions with China, for example).

The underlying logic is the standard cost-benefit reasoning. The RSM is something like a local benefit-cost ratio of reducing emissions, so countries with a higher ratio should want to do more.

I see two main issues with the study. First, there exist a number of cost-benefit models that look at the actual incentives faced by different regions/countries. Starting with Nordhaus and Yang (1996), a variety of cost benefit models have looked at what is referred to as an open look nash equilibrium to determine local incentives. The overwhelming consensus of these studies is that local incentives are largely insufficient to keep us below 3 degrees, let alone 2 degrees. One of the contributing factors to this is the free-rider problem that results when a number of different decision makers only look at their local costs and benefits in a problem with an externality. By using the local SCC, the authors focus on just such local incentives which are known to be insufficient. So even though the RSM might provide some ordinal information on how countries are ranked by the local cost-benefit ratio for a given scenario, this is not really useful in determining how and who should apply more ambition in mitigation. It is known that some degree of

cooperation amongst nations is necessary to bring ambition up and escape the free-rider problem.

My other issue with the analysis is that it is evaluating countries at the ambition levels determined by the SSP scenarios. They take emissions levels for each region for, for example SSP2-RCP4.5, and compute what the marginal abatement cost for each country would be at that emission level in the GTAP E model. The emission levels in the IIASA SSP database from which they take the scenarios are not necessarily related to actual emissions in the different countries, nor are they related to the NDCs. In fact, the distribution of emissions for each scenario in the database is simply what the IAM with which the scenario was calculated proposed. So if the EU is reducing its emissions significantly in RCP 4.5, it is because the AIM/CGE, GCAM4, REMIND-MAGPIE, IMAGE, or WITCH-GLOBOIM models optimise for it to be that way. When the authors then find that at these high level of mitigation the GTAP E marginal cost of mitigation in the EU is high, this is simply an artefact of the difference in implicit marginal abatement cost curves between GTAP E and these other IAMs. That is not very useful information on its own. For the reader to get something useful out of the analysis it would be good to provide a comparison of the ACTUAL marginal costs of abatement given actual emissions. There is nothing tying the optimal distribution of emissions in the SSP/RCP scenarios to what is actually going on. So to claim something about which countries have room for further action based on the scenario emissions without ensuring that these scenarios are close to reality is not very credible.

Furthermore, it is my understanding that when the WITCH model proposes a solution, it has each region apply the same marginal cost of abatement. So when for the same scenario GTAP E finds wildly different marginal costs of abatement, it would be edifying to find out why. From the authors description GTAP E models trade, while WITCH doesn't. If trade effects are the source of this difference, then that should be analyzed and emphasized. As presented one just gets a picture of the fact that GTAP E is different from the average of the 5 IAMs it is being compared to. More interesting would be to know WHY it is different, and whether the trade channel is what is making mitigation cheaper in high RSM countries and more expensive in low RSM countries. Or whether it is other idiosyncratic differences between GTAP E and the comparator IAMs.

With those two main critiques in mind I think that the analysis as it currently stands does not support the use of RSM as a tool to evaluate different countries' current mitigation actions. But with some modification, the underlying analysis can be made into a valuable contribution to the literature. But the focus would have to change into one that compares GTAP E to the SSP IAMs more directly, or one that looks at current or planned mitigation action, rather than IAM model outputs.

Nordhaus, William D., and Zili Yang. "A regional dynamic general-equilibrium model of alternative climate-change strategies." *The American Economic Review* (1996): 741-765.

REVIEWER COMMENTS

Reviewer #1 (Remarks to the Author):

This research creatively utilizes a simple yet effective indicator, the RSM, to answer an important question: how we should conduct carbon emission mitigation in the most cost-effective way under the international scope. The research question is in accordance with the “common but differentiated responsibilities” principle proposed by UNFCCC, thus bearing great significance in policy suggestions for international climate mitigation collaborations and negotiations. The manuscript itself is very well structured. The presentation of method is clearly organized and articulated, nicely facilitated with schematic diagram for readers’ quick understanding. Results are effectively illustrated in the figures produced, accompanied by insightful analysis to formulate an answer to the research question proposed in the beginning. Overall, this research manuscript is produced in very high quality. I believe it is almost ready to be scientifically published.

Re: Thank you for your helpful comments and suggestions, which have been incorporated in our revised manuscript.

One comment regards to the terminology. In the entire discussion, the authors seem to associate RCC only with carbon tax. However, it might be a too narrow definition for carbon pricing tools. The other most important tool for carbon pricing is emission trading scheme (ETS). In addition to setting a fixed tax rate for each tonne of carbon emitted, the same carbon price could also be achieved through “cap and trade” schemes. Hence, narrowing the discussion for carbon pricing to only carbon tax would not be the most precise approach. I would suggest the authors to consider broadening their discussion on carbon pricing to wider variety of policy tools.

Re: Thanks for your suggestion. RCC is used to measure the economic loss of carbon emission reduction thereby which is calculated by the loss of GDP dividing by the amount of carbon emission reduction. Carbon pricing tools includes carbon tax and carbon trade. This study adopts carbon tax to achieve carbon emissions reduction targets. To be specific, national emission reduction targets are obtained by downscaling the global SSP scenarios, using carbon tax to internalize the marginal cost of carbon emission reduction in various countries as well as through increasing the carbon tax to achieve the carbon emission reduction target, and then calculates the GDP loss under the SSP scenario. This method has also been widely used in existing research¹⁻³. In this study, we did not consider the mechanism of carbon trade despite it is used widely⁴⁻⁶. In terms of mechanism, there are similarities between carbon pricing and carbon tax. The mechanism of carbon trade is also to increase the marginal cost of carbon emission through endogenous carbon price to achieve the carbon emission reduction targets. The mechanism of carbon trade and carbon emissions reduction can cause nearly consistent results. However, the mechanism of carbon trade involves some complicated problems of carbon exchange quota allocation which could increase the uncertainties of results. Thus, in this study we adopt carbon tax tool to achieve carbon emissions reduction which will not distort the simulation results.

Please find the following my comments:

1. After reading the whole paper, I can understand your title, but it is not that straightforward what your research is about for a broader audience. It would be nice to rephrase it or clarify the regional carbon mitigation “suitability”.

Re: Thanks for your suggestion. Please check lines 99 to 101 in the revised version:

“The relative suitability for mitigation (RSM) is constructed as the ratio of normalized SCC to normalized RCC for each emitter”.

2. Intended Nationally Determined Contribution is usually abbreviated to INDC.

Re: Thanks for correction. The abbreviations of Intended Nationally Determined Contribution (INDC) and Nationally Determined Contribution (NDC) have been added separately. See lines 40 and 46.

3. Authors used GTAP-E model in this work. Many other IAMs, such as DICE/RICE, have also been used to discuss the similar topics (Yang et al., 2018, Mi et al., 2019). It will be better if authors add more explanations why GTAP-E was selected. What are its advantages?

Re: Thanks for your valuable comments. Many previous studies have calculated the cost of carbon emission mitigation by IAM models, including DICE, RICE, GCAM, etc. Compared with these models, GTAP-E model has two significant advantages: **1)** GTAP-E model covers more detailed regions and industrial sectors. In this study, 141 countries/regions in GTAP-E model are aggregated into 27 regions which provides the cost of carbon emission mitigation in more major carbon emitting countries around the world. In contrast, the existing research only includes the emitting countries like China, the US, European countries, etc., little estimating the cost of carbon emission reduction for other emitting countries. **2)** When calculating the cost of carbon emission mitigation, DICE, RICE and GCAM models mainly consider the negative correlation between regional carbon emission intensity and the cost of carbon emission mitigation. However, the regional cost of carbon emission mitigation also depends on the trade structure of the region. The impacts of carbon emissions reduction will alter the regional economic loss through the transmission and feedback of international trade in regions, and thus affecting RCC. Compared with most IAMs, GTAP-E model as a multi-regional CGE model can link countries together by interregional trade and depicts the economic impacts of carbon emissions reduction under the interregional mechanism of transmission and feedback.

In addition, we also compared the difference in evaluations of average RCC between our study and previous studies. These previous studies have estimated the average cost or marginal cost of carbon emissions mitigation by using DICE, EMF22, GCAM, EPPA and FAIR models⁷⁻¹² (see Table below). Although our results show there is difference in absolute value, the ranking of RCC in regions is consistent with the existing results. The developing countries such as China, India, etc. have the lower RCC, whereas some developed countries like European countries and the US have the higher RCC. In addition, to obtain more reliable estimation results, RCC is standardized in this

paper, which decreases the impact of model selection on the simulation results.

	1	2	2	3	4	5	6
Authors	Paltsev et al., 2007	Stern et al., 2011	Stern et al., 2011	Elzen et al., 2011	Li et al., 2015	Hof et al., 2017	Liu and Feng, 2018
Scenario	Reduce carbon emissions by 25% towards 2010	Reduce carbon emissions by 25% towards 2020	Reduce carbon emissions by 25% towards 2020	Realize the Copenhagen Accord By 2020	Reduce carbon emissions by 100 million tones CO ₂	Realize the NDC goals by 2030	The changing of carbon emission in 2000-2014
Index	Marginal cost of carbon reduction	Marginal cost of carbon reduction	Average cost of carbon reduction	Marginal cost of carbon reduction	Average cost of carbon reduction	Average cost of carbon reduction	Shadow Price
Model	EPPA	EMF22	EMF22	Fair	CGE	GCAM	DDF
Unit	USD/t	USD/t	USD/t	USD/t	USD/t	USD/t	USD/t
CHINA		42.38	25.31	4	43.62	4.37-10.15	159
EU	205	116.39	98.63	35-58	53.16	8.12-44.55	
India		37.58	7.4	0-1	47.82		589
US	231	70.23	128.81	41-42	57.27	10.64-27.07	314
Brazil				72-108		1.92-98.04	683
Canada	127			31-32		10.15-48.78	672
Indonesia				35-89		3.83-20.41	695
Japan	323			79-80	62.28	0-21.74	823
Mexico				4-273		14.71-80.00	690
Russia						5.95-23.81	550
France							760
Germany							730
Sweden							732
Korea				128			

4. Line 47: Better define the economic benefits here. I'm not sure if the "incomplete and uncertain information on the economic costs and benefits" are the primary barriers to promoting enhanced mitigation ambition. Every policy in the US needs to pass a cost-benefit test, yet they have no invitation to do as climate change is a public good.

Re: Thanks. We do understand our statement might be too strong and absolute. This sentence has been modified to (See lines 47 to 49 in the revised main text):

"Straightening up information on the economic costs and benefits of fulfilling such ambition could help a part country to make mitigation strategy and enhance mitigation ambition."

5. Line 60-63: I don't think international trade will affect the low-carbon technology cost for countries. Could you provide an example of how carbon reduction costs will be affected? Adding a definition for the carbon reduction cost can also be very helpful.

Re: Thanks for reviewer's suggestions. International trade does not affect the low-carbon technology cost directly. RCC is defined as the GDP loss dividing by carbon emissions reduction. Due to the difference of carbon emissions intensity and trade structure among countries, carbon emissions mitigation will alter the competitiveness of each region in the international market, and subsequently affect the import and export of each region, thus changing the GDP loss of countries. Even under the same technology cost, the transmission and feedback effect of trade will change the cost of carbon emission mitigation.

We supplemented the sensitivity experiment to explain the impact of interregional transmission and feedback mechanism on the RCC. In the test, we turned off the price transmission mechanism of GTAP-E model, making each region as a single-regional CGE model. In this case, the RCC in each region is not affected by the emissions reduction of other regions. The results of sensitivity experiment show that if the mechanism of interregional transmission and feedback is ignored, the

economic cost associated with RCC will be overestimated or underestimated depending on the countries. To be specific, for fossil energy importers such as China, the economic loss of carbon emission reduction will be overestimated. For China, when international impact of transmission and feedback mechanism is ignored, the average loss of GDP is 0.17%, and RCC is 8.03 USD/t CO₂; When considering the international impact of transmission and feedback mechanism, GDP increases by 0.16% on average and RCC is -7.33 USD/t CO₂. On the contrary, for energy exporting countries such as Russia, the economic loss of carbon emission reduction will be underestimated. For Russia, when without considering international impact of transmission and feedback mechanism, the average loss of GDP is 0.02%, and RCC is 49.92 USD/t CO₂; When considering the international impact of transmission and feedback mechanism, GDP increases by 0.24% on average and RCC is 3.08 USD/t CO₂. In sum, global carbon mitigation will reduce energy demand and energy price in a certain degree. When trade mechanism is considered in GTAP-E model, energy-importing regions will benefit from the lower energy price and expand domestic production. On the contrast, energy-exporting regions will be suffered from the lower energy export and reduced relative competitiveness in global market. See lines 132 to 142 in the revised main text:

“Here the sensitivity experiment was supplemented to explain the impact of interregional transmission and feedback mechanism on the average reduction cost of carbon (Supplementary Table 5). In the test, we turned off the price transmission mechanism of GTAP-E model, making each region as a single-regional CGE model. In this case, the average reduction cost of carbon in each region is not affected by the emissions reduction of other regions. The results of sensitivity experiment show that if the mechanism of interregional transmission and feedback is ignored, the economic cost associated with average reduction cost of carbon will be overestimated or underestimated. In sum, the effect of trade mechanism could be mainly explained by two major channels. On one hand, the global carbon mitigation will directly reduce energy demand and energy price in a certain degree. When the trade mechanism is motivated in GTAP-E model, energy-importing regions will benefit from the lower energy price and cut down the costs of their domestic production, alleviating the GDP losses caused by the carbon mitigation. But energy-exporting regions will experience greater GDP losses because of the decreasing revenues from energy exports. On the other hand, as energy-exporting countries mostly have the relatively high carbon intensity, carbon mitigation will raise their costs of domestic production more significantly, compared with energy-importing countries. As a result, energy-exporting countries will lose the comparative competitiveness in global market, aggravating GDP losses caused by carbon mitigation. ”

6. “The cost of carbon emission reduction” is an essential concept in this paper and I suggest using words rather than abbreviations to provide a fluent reading experience. Also, as the unit is US\$/tCO₂, should you use ‘unit cost of emission reduction’ or similar ones to avoid misunderstanding?

Re: Good suggestions. In the revised version, we avoid using abbreviations as you suggested for most descriptions in the main text. And we used “average reduction cost of carbon” instead of their previous names.

7. Reference 7 also investigates the benefit aspect of NDCs. How is your research different from theirs?

Re: Thanks for your question. Both our studies are aimed to help enhance climate action towards achieving the Paris goal and used the NDC to evaluate the mitigation target and ambition for each

region. For difference, Reference 7 (Yang et al) is focusing on the equity perspective with the reduction amount of NDC, and further compared with scenario based on solely economic emission level; Our study is focusing on the ambition perspective with mitigation ambition score of NDC, and further compared with scenario based on relative ranked suitability of mitigation.

In detail, the differences between our study and Reference 7 on the NDCs are: **1)** For the indicator, Yang's study focused on the mitigation target of NDC which was quantified to the reduction of emission for each region based all the national NDC Factsheet, and our study focused on the ambition score of NDC which was determined based on the degree of warming when the NDC of a given country been applied globally; **2)** For the perspective, although we both mentioned the data from Paris Equity Check, Yang's study focused on the data of multi-equity map which show an assessment of countries' climate pledges under five visions of climate justice and for emissions pathways, and our study focused on the data of pledged warming map which show an assessment of global warming when all countries follow the ambition of a given one; **3)** For the benefit aspect of NDCs, Yang's study compared the assumed amount of emission based on NDC mitigation target to emission based on solely economic emission level (when the marginal mitigation cost is equal to the marginal climate damage avoided at a national level) for each region under each SSP scenario; our study compared the relative rank based on NDC ambitions score to the relative rank suitability of mitigation (based on the ratio of normalized social cost of carbon and unit reduction cost of carbon) among all regions under each SSP-RCP mitigation scenario.

8. Line 68-71: The sentence is quite hard to read. What are the potential benefits of emission reduction? Add an example here. Reference 18-24 adopted different equations to calculate SCC. Some defined the SCC as welfare loss, while others define SCC as GDP loss. The SCC estimation is blooming in recent years. I suggest the authors refer to more updated SCC estimates under SSPs and justify why you are choosing Ricke's estimates.

Re: The potential benefit of emission reduction is considered as the avoided economic cost associate with avoided climate damage. For example, the social cost of carbon of China is US\$25.7 per tCO₂ (5.62-51.49) under SSP2-RCP4.5 Scenario, which represents that the potential benefit of emission reduction is US\$25.7 per tCO₂ (5.62-51.49) when the emission of an additional tonne of carbon dioxide avoided. To avoid misunderstanding, we updated the description here (see lines 72 to 76):

“The estimated RCC can be contrasted against the potential benefits of emission reduction. The potential benefits of emission reduction are considered as the avoided economic costs associate with avoided climate damage, whose economic value is defined here to be equivalent to the social cost of carbon (SCC, US\$ per tCO₂) for each metric tonne of CO₂ emissions that can be otherwise removed.”

And as you suggested, we have cited and referred to more updated SCC estimates under SSPs. The reason why we are choosing Ricke's estimates are: **1)** As the reviewer mentioned, basically SCC can be represented as the welfare loss¹³⁻¹⁸, or as GDP loss¹⁹⁻²². Considering that our RCC is defined as the as the potential GDP loss in a given country/region as a result of action to remove one metric tonne of CO₂ emissions, we choose the SCC defined as the GDP loss by Ricke et al so that these two indicators (RCC vs SCC) can be compared in an apple-to-apple magnitude. **2)** For most of SCC related studies, they are focused on global or some specific regions (disaggregate the global economy in up to 6–16 macro-regions^{13,16,17,19,21-23}), or on some specific scenario (basically for some

counterfactual scenarios or only focusing on SSP scenarios^{13,15,18}). Among all these studies, Ricke's results provided detailed country-level SCC under several combined SSP-RCP mitigation scenarios, which could provide more accuracy when mapping with our country-level RCC results and further be used to calculate the RSM for each country under multiple mitigation scenarios in this study.

Besides, we do realize that only including the SCC based on Ricket's results might not capture the variety of current SCC estimation. So we added some new SCC results by Yang et al based on the latest version of the RICE model with modifications and updates on climate model, regional definition and damage function¹⁸. Note that we mapped the Yang's results from 15 regions to our 27 regions here. See Supplementary Table 5 and Supplementary Figure 6 (as is shown below) for the detailed information. Considering that we are only focusing on relative rank rather than the exact value of SCC, all SCC values have been normalized here. Our results show that the RSM based on Ricket's SCC or Yang's SCC are both showing the similar distribution even with some difference for some countries. Our findings about the mismatch between suitability and ambition (Figure 3), are still robust and significant even with different kind of SCC. Because behind this phenomenon is the negative correlation between the global economic landscape and geographic distribution. See lines 266 to 274 for the descriptions:

“Third, the calculation of SCC follows Ricket et al., and is affected by the statistical method and functional forms used to assess economic damage, although the relative ranking of countries is robust under each scenario. Therefore, we have tested the robustness of our results by constructing RSM with SCC results by Yang et al based on the latest version of the RICE model with modifications and updates on climate model, regional definition and damage function (Supplementary Table 5). Our findings about the mismatch between suitability and ambition (Supplementary Figure 6) is still robust and significant even with different kind of SCC because that behind this phenomenon is the negative correlation between the global economic landscape and geographic distribution.”

Supplementary Figure 6: Comparison of RSM (relative suitability of mitigation) based on SCC (social cost of carbon) by Ricke et al and Yang et al. (a) Contrasting suitability and ambition of carbon mitigation among 27 emitting countries and regions with RSM constructed based on SCC by Ricke et al. (b) Contrasting suitability and ambition of carbon mitigation among 27 emitting countries and regions with RSM constructed based on SCC by Yang et al.

9. Figure 1: Better avoid abbreviation in figure titles.

Re: Done as suggested. All figure titles have been updated with explanations of abbreviations.

10. Figure 3: Why do the rest of Western Asia and the US have such a huge uncertainty range for SCC?

Re: One of the major sources of SCC's uncertainties is come from the climate system response to CO₂ (climate sensitivity) and thus higher CO₂ emissions tends to lead a higher uncertainty of SCC²⁴. Therefore, for regions like rest of Western Asia and the US with huge amount of CO₂ emissions would have a huge uncertainty range for SCC.

11. Figure 4: How to interpret these RSM numbers? Where is the benchmark for domestic mitigation/international mitigation transfer?

Re: RSM is calculated by the ratio of normalized SCC to normalized RCC for each emitter. Regions with higher RSM number represents with higher suitability (relative high SCC and low RCC), and vice versa. For example, among all regions, rest of western Asia and India tends to be the most “suitable” region to achieve the mitigation, and EU would the least “suitable” region to reduce the domestic emission; And among all scenarios, world under SSP3-RCP6.0 Scenario would be with most urgent to achieve carbon mitigation.

The number of RSM only represents the relative value of relationship among all regions under each mitigation scenario. As for the benchmark for domestic mitigation/international mitigation transfer, it is actually not directly related or shown in this Figure 4. The related hypothetical scenario with international mitigation transfer is designed based on Scenario SSP2-RCP4.5 with GTAP-E model, in which European Union, a region with low RSM and high ambition, transfers 10% of their mitigation amount (30.4 (28.3–40.6) million tCO₂) to China, a region with high RSM and low ambition (although with a recent ambitious pledge for carbon neutrality). See lines 234 to 249 in the main text:

“To further demonstrate the economic mutual benefits of cross-regional emission mitigation collaboration, we conduct a hypothetical experiment based on Scenario SSP2-RCP4.5 with GTAP-E model, in which European Union, a region with low RSM and high ambition, transfers 10% of their mitigation amount (30.4 (28.3–40.6) million tCO₂) to China, a region with high RSM and low ambition (although with a recent ambitious pledge for carbon neutrality 33,34). We find that the RCC of the European Union would decrease significantly from US\$ 97.2 per tCO₂ (89.7–114.7) to US\$ 90.1 per tCO₂ (82.5–105.7). For China, its RCC would only increase slightly from US\$ -0.8 per tCO₂ (-2.6–0.9) to US\$ 0.9 per tCO₂ (-0.7–2.2). Therefore, the double-win situation could be achieved through the Sustainable Development Mechanism 1, with necessary improvements, to support China’s carbon mitigation. This Sino-Europe collaboration would also avoid US\$ 4.65 (3.55–7.37) billions of GDP loss for the world through trade-associated inter-regional connections. If the transferred portion of emission mitigation increases to 50%, the avoided world GDP loss would increase to US\$ 20.2 (15.8 – 31.2) billion; and the average reduction cost of carbon would become US\$ 5.9 per tCO₂ (4.5–6.9) for China and US\$ 52.1 per tCO₂ (42.0–64.3) for the European Union. Country-specific results for GDP changes are shown in Supplementary Table 5.”

12. I suggest the authors separate the conclusion from the discussion and be more explicit about the findings and policy implications.

Re: Thanks for your suggestion. Now we separated the “discussion” section to two sections: “Uncertainty discussion” and “Conclusion”. Besides, we have updated the Conclusion part with more explicit about the findings and policy implications as you suggested, see lines 291 to 312:

“Today, clearly the momentum of the current mitigation schedule for the whole world does not appear to be sufficient to achieve the goal of keeping the temperature rise within 2°C or 1.5°C. How to improve the international cooperation rather than individual policies on climate change mitigation is crucial to achieving the Paris goal. Even a cross-regional policy framework has been proved as an effective solution considering large differences across regions to reach net-zero carbon emissions, how to build the cooperation for regions with cross-regional policy framework or global climate club is the major challenge. Besides, another major challenge is large disagreement about the benchmarks by which each country’s should enhance their ambition. This

study offers an RSM-based framework to help raise regional emission mitigation ambition and provide a guidance for mitigation cooperation from the economic perspective. More affordable and capable emitters with relatively high RSM but low NDC ambition, such as China and Thailand, might consider to substantially raise their mitigation ambition for their own economic and environmental interests. More affordable emitters with low RSM but high ambition, particularly the European Union and Switzerland, and less affordable developing countries with high RSM but low ambition might consider working collaboratively to reduce emissions and share credit of such action. Such cooperation would be more economically viable for both parties and is supported by the 6th Article of the Paris Agreement. For example, the cooperation between Norway and Indonesia based on reducing carbon emissions from deforestation and forest degradation has already shown contributions to tropical countries NDCs and significant emissions reduction²⁵. Together, enhancement of domestic and internationally collaborative mitigation action, aided by better knowledge on cost and benefit and thus enhanced ambition, will be crucial for successful climate change mitigation.”

References:

P Yang, YF Yao, Z Mi, YF Cao, H Liao, BY Yu, QM Liang, DM Coffman, YM Wei. Social cost of carbon under shared socioeconomic pathways. *Global Environmental Change*. 2018, 53: 225-32.
Z Mi, H Liao, D Coffman, YM Wei. Assessment of equity principles for international climate policy based on an integrated assessment model. *Natural Hazards*. 2019, 95, 309-323.

Re: Thanks for your suggestion. All references have been cited and discussed in the revised main text.

Reviewer #2 (Remarks to the Author):

This is a great paper and should be published by Nature Communications. I do not have any important reservations and so I recommend simple acceptance, but if other reviewers raise any important technical questions I urge the editors to give the authors generous opportunity to respond. Mindful of the impending IPCC deadline, if there are no objections that are legitimate raised by other reviewers, I urge the journal to consider a rapid acceptance.

This paper provides a novel comparison, for 27 nations, of the costs of mitigation to the benefits of avoided damages under policy-relevant mitigation scenarios for each of those nations, and to ambition in NDCs. The authors choose logical and defensible methods for doing this among those that are readily available, and explain and write their paper with admirable clarity -- clarity that far surpasses the average climate policy article in journals like Nature and Nature Climate Change. As with any such exercise, there are many ways in which a reader might quibble with one or more aspect of the chosen methods (and in particular, the models relied upon, and their parameter assumptions) -- but that is unavoidable, and the authors are forthright about this and the assumptions made, and again the authors have chosen logical, defensible, and policy-relevant methods currently available to make these important comparisons.

Mainly for this reason, I believe the paper should be published in this journal, and if any referees disagree based on quibbles about the methods, I urge the editors to critically examine whether such

quibbles actually point to technical worries about the implementation of those methods, or to important worries about correct choice of methods, since again any possible choice of methods will be subject to quibbles, and the authors seem to me to have made reasonable choices for deriving a 'central readily-available estimate' for these comparisons.

With the comparisons in hand, the authors wisely connect the dots between their results and how they can help inform pressing policy questions, such as how to form effective 'linkages', 'climate clubs', where to expect barriers to arise to global cooperation on GHG reductions, etc. They connect the dots to the underlying conceptual and empirical considerations: eg that what matters for mitigation opportunities is partly a matter of where it is cheap to make reductions, not merely where there are a lot of emissions, etc. The authors also provide clear and well-designed supplemental data summarizing the quantitative results and the numbers used to derive them that are themselves the outputs of models or other studies.

One idea, perhaps for the authors' future work, is to test the robustness of their central results to the use of other alternative models. For example, one could look at SCC values using other widely used multi-region SCC models, eg RICE, FUND, or PAGE. (Note that the Ricke damage function used by the authors is not used by eg US government estimates, and is based on controversial Burke-Hsiang-Miguel damage function.) Similarly, the authors could try alternative mitigation cost models (eg GCAM). The authors could test robustness for the subset of their 27 regions for which there is a direct comparison with regions within a chosen alternative SCC model and chosen alternative mitigation cost model. Because I assume that the basic qualitative story would remain the same given different model choices -- ie the same nations would have qualitatively similar match or mismatch between costs and benefits and ambition -- I do not demand that the authors perform these robustness checks, but it might add value, perhaps in future work. The authors could perhaps also add a sentence explaining the normalization method and the motivation for it just a slight bit more. The paper will be the basis for useful future work as well that builds on this -- eg evaluation of climate club possibilities.

Again, this is a great paper and should be published by Nature Communications. I do not have any important reservations and so I recommend simple acceptance, but if other reviewers raise any important technical questions I urge the editors to give the authors generous opportunity to respond. Thanks to the authors for this contribution.

Mark Budolfson

Re: Really thanks for your highly appreciation and meaningful suggestions! Based on your comments, we have updated the main text in the following part:

1. We tested the robustness of our central results to the use of other alternative models. We used the SCC simulated by RICE model to test the robust of our finding about mismatch between SCC and RCC and mismatch between suitability (cost and benefit) and ambition. And just as the reviewer assumed, the basic qualitative story remains the same even given different model choices. See lines 266 to 274, Supplementary Figure 6 (as is shown below), and our reply to reviewer 1:

“Third, the calculation of SCC follows Ricke et al., and is affected by the statistical method and functional forms used to assess economic damage, although the relative ranking of countries is robust under each scenario. Therefore, we have tested the robust of our results by constructing RSM

with SCC results by Yang et al based on the latest version of the RICE model with modifications and updates on climate model, regional definition and damage function (Supplementary Table 5). Our findings about the mismatch between suitability and ambition (Supplementary Figure 6) is still robust and significant even with different kind of SCC because that behind this phenomenon is the negative correlation between the global economic landscape and geographic distribution.”

Supplementary Figure 6: Comparison of RSM (relative suitability of mitigation) based on SCC (social cost of carbon) by Ricke et al and Yang et al. (a) Contrasting suitability and ambition of carbon mitigation among 27 emitting countries and regions with RSM constructed based on SCC by Ricke et al. (b) Contrasting suitability and ambition of carbon mitigation among 27 emitting countries and regions with RSM constructed based on SCC by Yang et al.

2. We added more sentences to explain the normalization method and the motivation for it. See lines 101 to 107:

“Considering the large uncertainties of the magnitude of mitigation cost and benefit but the generally robustness of relative distribution, all the values of respective SCC and RCC have been normalized to range between 0 and 1 from the lowest to highest. Besides, comparing the normalized values of RCC and SCC can better represent suitability (in terms of the benefit versus cost) among

all regions and all scenarios. In sum, the normalization based on the min-max method is conducted to cancel out the effect of systematic errors (for all emitters) in the absolute values of RCC and SCC.”

Reviewer #3 (Remarks to the Author):

The study develops the relative suitability for mitigation (RSM), as a metric to compare countries' relative potential for further mitigation at a number of SSP/RCP scenarios. For each country the RSM is the ratio of the local social cost of carbon (SCC, defined as the discounted sum of marginal damages to the country from a ton of emitted CO₂) to the local marginal cost of reducing a ton of emissions at the respective scenario. The local SCC is taken from the literature (Ricke et al) and the marginal cost is computed by implementing the particular scenarios into GTAP E, an integrated assessment model (IAM) with the distinction of capturing trade effects.

It is important to understand that the metric evaluates the suitability for FURTHER abatement beyond the particular scenario in question. The evaluation does not depend on current emissions or even on the nationally determined contributions.

The authors find that there is large variation in RSM across regions and countries, and suggest countries with high RSM might have incentives to do more than under the given scenarios and those with low RSM might choose to trade mitigation effort with high RSM countries (they suggest the EU exchange it's reductions with China, for example).

The underlying logic is the standard cost-benefit reasoning. The RSM is something like a local benefit-cost ratio of reducing emissions, so countries with a higher ratio should want to do more.

Re: Thanks for your comments. And a little bit of supplement here: large variation in RSM across regions and countries is one of our findings. Another important finding is the large mismatch between RSM and NDC across regions. Therefore, what we want to highlight is not only the cooperation should be happened between high RSM countries and low RSM countries, but should also consider the variety mitigation ambition across countries. Cost and benefit and ambition are the three key points to define the further abatement and evaluation of climate club possibilities for each region.

I see two main issues with the study. First, there exist a number of cost-benefit models that look at the actual incentives faced by different regions/countries. Starting with Nordhaus and Yang (1996), a variety of cost benefit models have looked at what is referred to as an open look nash equilibrium to determine local incentives. The overwhelming consensus of these studies is that local incentives are largely insufficient to keep us below 3 degrees, let alone 2 degrees. One of the contributing factors to this is the free-rider problem that results when a number of different decision makers only look at their local costs and benefits in a problem with an externality. By using the local SCC, the authors focus on just such local incentives which are known to be insufficient. So even though the RSM might provide some ordinal information on how countries are ranked by the local cost-benefit ratio for a given scenario, this is not really useful in determining how and who should apply more ambition in mitigation. It is known that some degree of cooperation amongst nations is necessary to bring ambition up and escape the free-rider problem.

Re: Thanks for this comment. We have to admit that this study evaluated the feasibility of the carbon reduction for countries according to their local incentives. The carbon reduction is determined according to each country's cost and benefit, which is regarded as a Nash equilibrium²⁶. Even the carbon reduction based on the RSM index could achieve the social welfare optimum of each country, it is unable to ensure that the global welfare optimization is reached. Because the problems of the externality and free rider are not addressed. Theoretically, the allocation of carbon reduction across countries could be determined by optimizing a global social welfare function and achieving the Lindal equilibrium^{27,28}. An example is Yang²⁸, who proved that the solution of a social planner's problem with the social welfare weights proportional to the inverse of the private shadow prices of externalities is the Lindal equilibrium without transfers, achieving the social welfare optimum.

However, we consider the great difference between the reality and ideality. Because the authority at the global level is missing, the theoretically-optimal carbon reduction could not be realized easily. In reality, countries still adopted the INDC mechanism to commit their carbon reduction goal, which is determined by their own costs and benefits of carbon reduction. Aiming to improve the current INDC mechanism, this study evaluated the feasibility of carbon reduction scheme and found that the INDC contribution and RSM index are mismatched for most countries. The carbon reduction scheme based on the RSM index could improve the global welfare effectively, relative to the current INDC mechanism. For example, as several countries that have high RSM had raised relatively low NDC goals, the more ambitious carbon reduction of these countries could improve the global welfare. Although the carbon reduction based on the RSM index is unable to achieve the global welfare optimum, it could be regarded as a second-best scheme of carbon reduction, because the global welfare is improved from the current INDC mechanism. See lines 280 to 287 in the revised version:

“Sixth, the allocation of carbon reduction across countries could be determined by optimizing a global social welfare function and achieving the Lindal equilibrium theoretically, but the carbon reduction used here is determined according to each country's cost and benefit, which is regarded as a Nash equilibrium. Therefore, even the carbon reduction based on the RSM index could achieve the social welfare optimum of each country, it is unable to ensure that the global welfare optimization is reached. However, it still could be regarded as a second-best scheme of carbon reduction because the global welfare is improved from the current INDC mechanism.”

My other issue with the analysis is that it is evaluating countries at the ambition levels determined by the SSP scenarios. They take emissions levels for each region for, for example SSP2-RCP4.5, and compute what the marginal abatement cost for each country would be at that emission level in the GTAP E model. The emission levels in the IIASA SSP database from which they take the scenarios are not necessarily related to actual emissions in the different countries, nor are they related to the NDCs. In fact, the distribution of emissions for each scenario in the database is simply what the IAM with which the scenario was calculated proposed. So if the EU is reducing its emissions significantly in RCP 4.5, it is because the AIM/CGE, GCAM4, REMIND-MAGPIE, IMAGE, or WITCH-GLOBOIM models optimise for it to be that way. When the authors then find that at these high level of mitigation the GTAP E marginal cost of mitigation in the EU is high, this is simply an artefact of the difference in implicit marginal abatement cost curves between GTAP E and these other IAMs. That is not very useful information on its own. For the reader to get something useful out of the analysis it would be good to provide a comparison of the ACTUAL marginal costs

of abatement given actual emissions. There is nothing tying the optimal distribution of emissions in the SSP/RCP scenarios to what is actually going on. So to claim something about which countries have room for further action based on the scenario emissions without ensuring that these scenarios are close to reality is not very credible.

Re: Thanks for your comments. SSP scenarios are not directly represent the real world, but they are being considered as representing some kinds of possibility for the future. In other words, the real world would be covered among all SSP scenarios. And we do understand simulating results under scenario close to reality would be the best to help discussing the further action for each region. That is the reason why we used the SSP2-RCP4.5 mitigation scenario (which is assumed to be the most similar scenario to the real-world developing pathway) as the typical scenario to discussed in the main text. And what we want to say is that our findings are not only robust under this SSP2-RCP4.5 scenario, but under all possible mitigation scenarios from SSP1-5 and RCP4.5-Baselines. Comparison of RSM and ambition under each scenarios was provided in Supplementary Figure 7 of the revised version. Besides to avoid the worries about accuracy of the only used average emissions from five IAMs, we also provided the comparison of SCC and RCC based on each IAM model in revised version (Supplementary Figure3b-f).

In sum, the difference between hypothetical scenario and reality is always a big challenge for scenario analysis and strategy design. We did our best to show that our findings are robust under these SSP mitigation scenarios. Even if we could not exactly represent what will happen in the reality of future, at least the negative correlation between the global economic landscape and geographic distribution is captured and could help to evaluate the cooperation possibility. See lines 250 to 261 in the revised main text:

“First, the emissions under each scenario are averaged over simulation results from five IAM models. Even SSP scenarios are not directly represent the real world, they are being considered as representing some kinds of possibility for the future. In other words, the real world would be covered among all SSP scenarios. As is shown in Supplementary figure 3a, our findings are not only robust under this SSP2-RCP4.5 scenario, but under all possible mitigation scenarios from SSP1-5 and RCP4.5-Baselines. The comparison of RSM and ambition under each mitigation scenario is provided in Supplementary Figure 7. And although the accuracy of each model is subject to errors in model parameters and assumptions, the multi-model averaging reduces the influence of errors in individual models. To avoid the worries about accuracy of the only used average emissions from five IAMs, we also provided the comparison of SCC and RCC based on each IAM model in Supplementary Figure 3b-f.”

Supplementary Figure 7: Comparison of RSM (relative suitability of mitigation) and NDC (national determined contribution) ambition for each region under each mitigation scenario. For both RSM and ambition scores, “High” represents the top 1/3 among the 27 regions, “Medium” represents the middle 1/3, and “Low” represents the bottom 1/3.

Supplementary Figure 3: Contrasting regional RCC (average reduction cost of carbon) and SCC (social cost of carbon) under each mitigation scenario. (a) Results are averaged over five IAMs. (b-f) Results are based on each IAMs. The dashed lines indicate linear fitting.

Furthermore, it is my understanding that when the WITCH model proposes a solution, it has each region apply the same marginal cost of abatement. So when for the same scenario GTAP E finds wildly different marginal costs of abatement, it would be edifying to find out why. From the authors description GTAP E models trade, while WITCH doesn't. If trade effects are the source of this difference, then that should be analyzed and emphasized. As presented one just gets a picture of the fact that GTAP E is different from the average of the 5 IAMs it is being compared to. More interesting would be to know WHY it is different, and whether the trade channel is what is making mitigation cheaper in high RSM countries and more expensive in low RSM countries. Or whether it is other idiosyncratic differences between GTAP E and the comparator IAMs.

Re: Thanks for your question. The major difference is that compared with most IAMs, GTAP-E model as a multi-regional CGE model can link countries together by interregional trade and depicts the economic impacts of carbon emissions reduction under the interregional mechanism of transmission and feedback. We supplemented the sensitivity experiment to explain the impact of interregional transmission and feedback mechanism on the RCC. In the test, we turned off the price transmission mechanism of GTAP-E model, making each region as a single-regional CGE model. In this case, the RCC in each region is not affected by the emissions reduction of other regions. The results of sensitivity experiment show that if the mechanism of interregional transmission and feedback is ignored, the economic cost associated with RCC will be overestimated or underestimated. To be specific, for fossil energy importers such as China, the economic loss of carbon emission reduction will be overestimated. For China, when international impact of transmission and feedback mechanism is ignored, the average loss of GDP is 0.17%, and RCC is 8.03 USD/t CO₂; When considering the international impact of transmission and feedback mechanism, GDP increases by 0.16% on average and RCC is -7.33 USD/t CO₂. On the contrary, for energy exporting countries such as Russia, the economic loss of carbon emission reduction will

be underestimated. For Russia, when without considering international impact of transmission and feedback mechanism, the average loss of GDP is 0.02%, and RCC is 49.92 USD/t CO₂; When considering the international impact of transmission and feedback mechanism, GDP increases by 0.24% on average and RCC is 3.08 USD/t CO₂. In sum, global carbon mitigation will reduce energy demand and energy price in a certain degree. When trade mechanism is considered in GTAP-E model, energy-importing regions will benefit from the lower energy price and expand domestic production. On the contrast, energy-exporting regions will be suffered from the lower energy export and reduced relative competitiveness in global market. See lines 132 to 142 in the revised main text:

“Here the sensitivity experiment was supplemented to explain the impact of interregional transmission and feedback mechanism on the average reduction cost of carbon (Supplementary Table 5). In the test, we turned off the price transmission mechanism of GTAP-E model, making each region as a single-regional CGE model. In this case, the average reduction cost of carbon in each region is not affected by the emissions reduction of other regions. The results of sensitivity experiment show that if the mechanism of interregional transmission and feedback is ignored, the economic cost associated with average reduction cost of carbon will be overestimated or underestimated. In sum, the effect of trade mechanism could be mainly explained by two major channels. On one hand, the global carbon mitigation will directly reduce energy demand and energy price in a certain degree. When the trade mechanism is motivated in GTAP-E model, energy-importing regions will benefit from the lower energy price and cut down the costs of their domestic production, alleviating the GDP losses caused by the carbon mitigation. But energy-exporting regions will experience greater GDP losses because of the decreasing revenues from energy exports. On the other hand, as energy-exporting countries mostly have the relatively high carbon intensity, carbon mitigation will raise their costs of domestic production more significantly, compared with energy-importing countries. As a result, energy-exporting countries will lose the comparative competitiveness in global market, aggravating GDP losses caused by carbon mitigation.”

With those two main critiques in mind I think that the analysis as it currently stands does not support the use of RSM as a tool to evaluate different countries' current mitigation actions. But with some modification, the underlying analysis can be made into a valuable contribution to the literature. But the focus would have to change into one that compares GTAP E to the SSP IAMs more directly, or one that looks at current of planned mitigation action, rather than IAM model outputs.

Re: Thanks for suggestion. We believe the revised version could makes the contribution of our study much clearer now. Based on the reviewer's comments and suggestion, basically we have: **1)** explained the significance of our study as a second-best scheme of carbon reduction considering the ongoing NDC mechanism of the real world; **2)** explained the robust of our findings under all hypothetical scenarios which could provide reference to evaluate the cooperation possibility and strategy in reality; **3)** added the explanation about the mechanism of how trade impacts the mitigation cost for each region in GTAP-E model comparing with other IAMs.

Nordhaus, William D., and Zili Yang. "A regional dynamic general-equilibrium model of alternative climate-change strategies." *The American Economic Review* (1996): 741-765.

Re: Thanks. We have added and discussed some new related references as reviewer suggested in the revised version.

- 1 Budolfson, M. *et al.* Protecting the poor with a carbon tax and equal per
capita dividend. *Nature Climate Change* **11**, 1025–1026, doi:10.1038/s41558-021-
01228-x (2021).
- 2 Fujimori, S. *et al.* Land-based climate change mitigation measures can affect
agricultural markets and food security. *Nature Food* **3**, 110–121,
doi:10.1038/s43016-022-00464-4 (2022).
- 3 Soergel, B. *et al.* Combining ambitious climate policies with efforts to
eradicate poverty. *Nature Communications* **12**, 2342, doi:10.1038/s41467-021-
22315-9 (2021).
- 4 Bednar, J. *et al.* Operationalizing the net-negative carbon economy. *Nature*
596, 377–383, doi:10.1038/s41586-021-03723-9 (2021).
- 5 Gallagher, K. S., Zhang, F., Orvis, R., Rissman, J. & Liu, Q. Assessing the
Policy gaps for achieving China’s climate targets in the Paris Agreement.
Nature Communications **10**, 1256, doi:10.1038/s41467-019-09159-0 (2019).
- 6 Liu, Y., Tan, X.-J., Yu, Y. & Qi, S.-Z. Assessment of impacts of Hubei Pilot
emission trading schemes in China - A CGE-analysis using TermCO2 model.
Applied Energy **189**, 762–769,
doi:<https://doi.org/10.1016/j.apenergy.2016.05.085> (2017).
- 7 Stern, D. I., Pezzey, J. C. & Lambie, N. R. Where in the world is it cheapest
to cut carbon emissions? *Australian Journal of Agricultural and Resource*
Economics **56**, 315–331 (2012).
- 8 Hof, A. F. *et al.* Global and regional abatement costs of Nationally Determined
Contributions (NDCs) and of enhanced action to levels well below 2 C and 1.5
C. *Environmental Science & Policy* **71**, 30–40 (2017).
- 9 Paltsev, S., Reilly, J., Jacoby, H. & Tay, K. H. 282–293 (Cambridge
University Press, forthcoming in October, 2007).
- 10 Liu, J.-Y. & Feng, C. Marginal abatement costs of carbon dioxide emissions
and its influencing factors: A global perspective. *Journal of Cleaner*
Production **170**, 1433–1450 (2018).
- 11 Den Elzen, M. G. *et al.* The Copenhagen Accord: abatement costs and carbon
prices resulting from the submissions. *environmental science & policy* **14**, 28–
39 (2011).
- 12 Li, A., Zhang, Z. & Zhang, A. Why are there large differences in performances
when the same carbon emission reductions are achieved in different countries?
Journal of Cleaner Production **103**, 309–318 (2015).
- 13 Gazzotti, P. *et al.* Persistent inequality in economically optimal climate
policies. *Nature Communications* **12**, 3421, doi:10.1038/s41467-021-23613-y
(2021).
- 14 Dietz, S., Rising, J., Stoerk, T. & Wagner, G. Economic impacts of tipping
points in the climate system. *Proceedings of the National Academy of Sciences*
118, e2103081118, doi:10.1073/pnas.2103081118 (2021).

- 15 Tol, R. S. J. A social cost of carbon for (almost) every country. *Energy Economics* **83**, 555–566, doi:<https://doi.org/10.1016/j.eneco.2019.07.006> (2019).
- 16 Yang, P. *et al.* Solely economic mitigation strategy suggests upward revision of nationally determined contributions. *One Earth* **4**, 1150–1162, doi:10.1016/j.oneear.2021.07.005 (2021).
- 17 Yang, P. *et al.* Social cost of carbon under shared socioeconomic pathways. *Global Environmental Change* **53**, 225–232, doi:<https://doi.org/10.1016/j.gloenvcha.2018.10.001> (2018).
- 18 Yang, P. *et al.* The impact of climate risk valuation on the regional mitigation strategies. *Journal of Cleaner Production* **313**, 127786, doi:<https://doi.org/10.1016/j.jclepro.2021.127786> (2021).
- 19 Chen, Y., Liu, A. & Cheng, X. Quantifying economic impacts of climate change under nine future emission scenarios within CMIP6. *Science of The Total Environment* **703**, 134950, doi:<https://doi.org/10.1016/j.scitotenv.2019.134950> (2020).
- 20 Rode, A. *et al.* Estimating a social cost of carbon for global energy consumption. *Nature* **598**, 308–314, doi:10.1038/s41586-021-03883-8 (2021).
- 21 Russell, A. R., van Kooten, G. C., Izett, J. G. & Eiswerth, M. E. Damage Functions and the Social Cost of Carbon: Addressing Uncertainty in Estimating the Economic Consequences of Mitigating Climate Change. *Environmental Management*, doi:10.1007/s00267-022-01608-9 (2022).
- 22 Kikstra, J. S. *et al.* The social cost of carbon dioxide under climate–economy feedbacks and temperature variability. *Environmental Research Letters* **16**, 094037, doi:10.1088/1748-9326/ac1d0b (2021).
- 23 John, W. Some Contributions of Integrated Assessment Models of Global Climate Change. *Review of Environmental Economics & Policy*, 115–137 (2017).
- 24 Ricke, K., Drouet, L., Caldeira, K. & Tavoni, M. Country-level social cost of carbon. *Nature Climate Change* **8**, 895–900, doi:10.1038/s41558-018-0282-y (2018).
- 25 Groom, B., Palmer, C. & Sileci, L. Carbon emissions reductions from Indonesia’s moratorium on forest concessions are cost-effective yet contribute little to Paris pledges. *Proceedings of the National Academy of Sciences* **119**, e2102613119, doi:doi:10.1073/pnas.2102613119 (2022).
- 26 Nordhaus, W. D. & Yang, Z. A regional dynamic general-equilibrium model of alternative climate-change strategies. *The American Economic Review*, 741–765 (1996).
- 27 Yang, Z. Identifying The Lindahl Equilibrium Without Transfers As A Social Optimum. *Metroeconomica* **64**, 25–43 (2013).
- 28 Yang, Z. *The Environment and Externality: Theory, Algorithms and Applications*. (Cambridge University Press, 2020).

Reviewer comments, second round

Reviewer #1 (Remarks to the Author):

My comments are responded well.

Reviewer #2 (Remarks to the Author):

This is a great paper and should be published by Nature Communications. The authors provide careful and compelling responses to the helpful comments from other reviewers. The authors choose logical and defensible methods, and explain and write their paper with admirable clarity -- clarity that far surpasses the average climate policy article in journals like Nature and Nature Climate Change. Thanks to the authors for their contribution.

REVIEWER COMMENTS

Reviewer #1 (Remarks to the Author):

My comments are responded well.

Re: Thank you for the helpful comments and suggestions.

Reviewer #2 (Remarks to the Author):

This is a great paper and should be published by Nature Communications. The authors provide careful and compelling responses to the helpful comments from other reviewers. The authors choose logical and defensible methods, and explain and write their paper with admirable clarity -- clarity that far surpasses the average climate policy article in journals like Nature and Nature Climate Change. Thanks to the authors for their contribution.

Re: Thanks for your positive feedback and appreciation.